# Virtual Home Assistant Use and Perceptions of Usefulness by Older Adults and Support Person Dyads

**DOI:** 10.3390/ijerph18031113

**Published:** 2021-01-27

**Authors:** Cynthia F. Corbett, Elizabeth M. Combs, Pamela J. Wright, Otis L. Owens, Isabel Stringfellow, Thien Nguyen, Catherine R. Van Son

**Affiliations:** 1College of Nursing, University of South Carolina, Columbia, SC 29208, USA; combsel@email.sc.edu (E.M.C.); shealypw@email.sc.edu (P.J.W.); stringi@email.sc.edu (I.S.); nguyentp@email.sc.edu (T.N.); 2College of Social Work, University of South Carolina, Columbia, SC 29208, USA; owenso@mailbox.sc.edu; 3College of Nursing, Washington State University, Pullman, WA 98686, USA; vansonc@wsu.edu

**Keywords:** virtual home assistants, digital home assistants, aging in place, caregivers, virtual assistant, assistive technology, digital devices, intelligent virtual assistants, smart home device, independent living, low cost

## Abstract

Aim: Describe virtual home assistant use and usefulness from the perspective of older adults and their support persons. Methods: This was a mixed-methods study with older adults and their support persons (*n* = 10 dyads). Virtual home assistant (VHA) equipment was installed in participants’ homes, and its use was documented for 60 days. Participants received protocol-guided telephone calls to address their VHA questions or problems. The type and frequency of VHA use were summarized with descriptive statistics. End-of-study interviews about VHA use were conducted with dyad participants. Qualitative content analyses were used to describe the interview findings about the dyad’s perceptions of using the VHA, how it was used, any difficulties experienced, and suggestions for future VHA uses. Results: Participant dyads reported positive VHA perceptions, including the potential for VHAs to promote aging in place. Participants discussed the challenges learning the technology and replacing old habits with new ones. Participants offered recommendations for future VHA skills and for more education and training about using the VHA. Conclusions: The study findings suggest that VHAs may be useful for older adults as they age in place and offer reassurance for support persons.

## 1. Introduction

Worldwide, the population is aging, with a concomitant increase in people who have two or more comorbid conditions [1]. During the next two decades, the number of people in the United States over age 70 will increase by 90%, and the 80 years and older population will more than double [2]. Among people over age 65, about 24% have one chronic condition, and nearly 64% have two or more chronic conditions [3]. Older adults prefer to remain in their own homes, a concept known as “aging in place” [4]. Harnessing information and communication technology to sustain older adults’ abilities to age in place is critical to supporting their independence and for balancing health and social care resources [5,6,7,8,9]. Technology for facilitating aging in place among older adults exists for many uses, including social connectedness; safety; independent living; and entertainment [6,8] (e.g., smartphones, personal alarms, and tablet computers), but barriers to use them persist [9]. Technology use barriers include the age-related digital divide, cost, stigma, perceived lack of need for technology, concerns about privacy and trust, and general lack of technological literacy [5,10]. For example, smartphones are used by 62% of adults aged 70 and older; however, they are less likely than younger cohorts to report using them on a daily basis, indicating a potential lack of literacy [11]. However, emerging technology, particularly voice-activated technology that is commercially available at a low cost, may foster older adults’ abilities to manage chronic conditions [12,13,14] and support independent living [13,14,15,16,17] and may reduce actual or perceived caregiver burdens [14,16]. Research findings suggest that many older adults are interested in learning about and using technology to promote their independence, function, and ability to age in place [5,12,16,17].

The effects of using commercially available, low-cost, virtual home assistants (VHAs; e.g., Amazon Echo “Alexa” and Google Home) to support aging in place have received little attention [12,15,16,17]. Current VHAs have a wide range of voice-activated capabilities enabling assistance with the functions of independent living, communication, and entertainment. VHAs work through access to the internet and can be used to access and control other devices, such as smartphones, TVs, lights, alarm systems, and other “smart” technology [18]. These capabilities may be particularly beneficial to older adults with chronic conditions [16]. Researchers in the United Kingdom placed VHAs in the residences of 30 older adults with chronic conditions. Preliminary results indicated that the VHAs promoted independent living [12], mental health [12], and chronic condition management behaviors such as medication adherence [13]. Chronic conditions require management and are often associated with functional impairments, leading to the need for support from family and friends (i.e., informal/formal caregivers) to enable continued aging in place [19,20]. For example, caregivers are highly influential to the medication adherence of older adults, which can control the debilitating effects of the disease and prevent further degradation of their health [21]. Other forms of caregiver support such as grocery delivery and assistance with home maintenance also enable older adults to age in place [22].

Older adults’ needs for caregiving support increase with age [23,24], and their support persons are beginning to explore the use of virtual home assistants (VHA) for assistance [15,16]. Pradhan and colleagues [15] and O’Brien and colleagues [16] retrospectively evaluated reviews posted by those with a validated purchase of Amazon Echo. Pradhan et al. evaluated 346 reviews that described VHA use by someone with a cognitive, sensory, or physical disability. O’Brien and colleagues filtered reviews by key words to evaluate those who mentioned some type of “health” benefit. Both research teams reported that family members who cared for older adults and those with disabilities often commented that the VHA assisted them in their caregiving role. However, no research exists to date that specifically explores the use and potential usefulness of VHAs among older adults and the primary person identified as supporting them to live independently (i.e., older adult/support person dyads). The purpose of this feasibility study was to prospectively investigate the use and usefulness of VHAs among older adults and their designated support person dyads. Specifically, we: (1) described the types of use and the frequency by which older adults and their support persons used VHAs and (2) explored the perceptions of older adults and their support persons regarding the uses and usefulness of VHAs, including facilitators and barriers to use. 

## 2. Materials and Methods

### 2.1. Design

The research design included both quantitative and qualitative data collection, employed simultaneously and with equal status. The purpose of the quantitative component was to quantify the actual usage of the VHA technology, whereas the purposes of the qualitative component were to corroborate the quantitative evidence and add depth and scope to the findings based on participants’ perceptions following use of the VHAs. Findings from both types of data were integrated during the analysis. Thus, we used a concurrent triangulation design for this mixed-methods study [25].

### 2.2. Recruitment

Advertisements and announcements were placed at community centers, a faith organization, and a primary care clinic to invite adults, aged 70 and older, with two or more chronic conditions and lived within 40 miles of a large urban university campus in the Southeastern US. Adults who were at least 70 years of age were selected for inclusion for several reasons. We wanted to evaluate the use of VHA technology among people who were likely to be out of the work force and spent more time in their homes. Most older adults in the United States are eligible for retirement benefits at the age of 65; thus, 65 is often used to designate someone as an older adult. However, many adults in the United States continue to work past age 65 for a variety of reasons, including differential retirement incomes for those who retire at a later age. Second, we were interested in evaluating the possible usefulness of VHAs among older adult/support person dyads whereby the older adult required some current assistance from support persons. Functional disabilities have been shown to increase with age [23,24,26], and in one study, over 50% of those over age 65 required support with at least one instrumental activity of daily living, which are tasks such as shopping and managing finances [26]. Thus, we recruited those who were aged 70 years or older and could identify a support person who they were in contact with at least twice per week who was also willing to participate in the study. Additional inclusion criteria were that the support person lived in a home separate from the older adult and within 40 miles of the same university campus. Exclusion criteria were the current use of a VHA, cognitive impairment, severe hearing loss, and inability to speak and/or read English by either member of the dyad. 

### 2.3. Ethical Considerations

The study was approved for human participation by a multi-institutional review board (IRB). Older adults who expressed interest in the study were informed of the voluntary nature of participation and the study procedures, including the potential risks and benefits. Often, potential participants were given copies of the older adult and support person consent forms to share with their family members and/or primary support person prior to deciding whether to participate. We sought to recruit a diverse sample and budgeted to provide smartphones or internet services to participants if the potential participants did not have access to one or both, which are required for VHA use. Privacy issues related to VHAs “always listening” were explicitly discussed with all participants. Further, participants were provided detailed information about the process the researchers would use to collect data about their VHA use. Participant dyads who expressed continued interest in participating provided individual written informed consent and received copies of their respective consent forms.

### 2.4. Measures

At baseline, the demographic information was collected from the older adult participants, including age, sex, race, ethnicity, educational attainment, income level, and number of chronic conditions. Older adult participants were asked to describe the assistance provided by their support persons and the types and frequency of their communications and their use of information technology (e.g., mobile phones and computers). Older adult participants completed the Patient Reported Outcomes Measurement Information System (PROMIS) health and well-being questionnaires. The PROMIS Global Health Scale, version 1.2, is a psychometrically sound measure of physical, mental, and social health with 39 Likert-response questions and with alpha scores of 0.81 and 0.86 from Global Physical and Global Mental, respectively [27]. The Personal Well-being Index (PWI) [28] and the World Health Organization-5 (WHO-5) Well-Being Index [29] both have Likert response items and acceptable psychometric properties, with the PWI having an alpha score between 0.70 and 0.85 and the WHO-5 Well-Being Index having an alpha score of about 0.85. Support persons completed a demographic questionnaire and the Macera Caregiver Burden Scale, which has an alpha score of 0.87 [30].

The type and frequency of VHA use was tracked through the Alexa application (app) on the study-dedicated smartphone, which was used to set up each participant’s Amazon Echo device(s) (i.e., “Alexa”). The app automatically recorded the initial VHA commands. For example, if the participant said, “Alexa, play classical music”, the research team documented the command into the appropriate category (i.e., music). Importantly, when participants made voice or video calls, only the commands were recorded by the VHA app. The actual voice or video conversations were not recorded by the app. Participant uses recorded in the app were identified only by participant number. We transferred the de-identified data from the app to a spreadsheet on a weekly basis and documented the types of uses separately for older adult participants and support person participants. 

At the end of the study, qualitative interviews were conducted independently with older adults and support persons to obtain descriptive data about how they used the device and its everyday usefulness. Older adults were asked questions such as: “Share your experience using Alexa (i.e., Amazon Echo devices). How are you using Alexa daily? Describe how you use Alexa with (name of identified support person). Describe the benefits you have identified from using Alexa. Describe the negative experiences from using Alexa.” Support persons were asked questions such as: “Describe any positive affects you found when using Alexa in your caregiver role. If you could instruct Alexa to help you in your caregiver role, what activities might you include?” The goal of this data was, in part, to learn about changes that should be made for a larger study.

### 2.5. Study Procedures

#### Protocol

After scheduling via telephone, the first visit to the older adults’ homes included written informed consent and completion of the baseline study measures: demographics and the PROMIS Global Health Scale version 1.2. If the support person was present, they also completed written informed consent and their baseline study measures: demographics and the Caregiver Burden Scale. If the support person was not present, a separate visit was scheduled to their home to obtain their informed consent and baseline study measures. Home visits were then scheduled separately with each member of the dyad to install the equipment. Amazon Echo and Google Home function very similarly; however, Echo devices were selected for this study, because they saturate 70% of the market and are expected to maintain market share in the future [31]. The Amazon Echo devices were then set up in participants’ homes using a study-dedicated smartphone. Older adult participants received a second-generation Echo Show that had a 10.1-inch smart video screen and a third-generation Echo Dot smart speaker that was 3.9 inches in diameter and 1.7 inches high. Support persons received an Echo Spot that was 4.1 inches in diameter, 3.8 inches tall, and had a small video screen and smart speaker. Participants selected the location(s) to place the Echo device(s) in their homes. The teaching protocol focused on the basic VHA capabilities, such as information retrieval (e.g., weather and news), listening to music, and video calls. Teaching was initiated at the first home visit for all participants. Approximately one week later, older adults received a follow-up visit, and support persons received a telephone call to answer any questions they had about using the Echo devices and to provide further teaching, which was individualized according to their questions and needs. Thereafter, we contacted older adults and support persons via telephone once every two weeks to ask if they had any questions about using the Echo devices. 

After 60 days, a visit was scheduled with each participant to conduct interviews about their use of the Echo devices and about their perspectives on its usefulness, including their ideas about possible other ways VHAs could be used to promote aging in place. Following the onset of the Coronavirus Disease 2019 (COVID-19) pandemic in the US, phone calls were scheduled to complete the final study procedures rather than home visits. Interviews were conducted by research members, recorded on two separate devices, and then digitally transferred to a professional transcription service. The digital transfer occurred over an encrypted, health insurance portability and accountability act (HIPAA)-compliant server. Participants were able to keep the Echo devices at the end of the study. 

### 2.6. Analyses 

Quantitative data was managed and analyzed using the statistical program for social sciences (SPSS 26.0, IBM, Chicago, IL, USA). Descriptive techniques were used to summarize participants’ demographic and health-related characteristics and the data collected from the Alexa app on VHA use. Specifically, frequencies (e.g., sex) or means (e.g., age) were computed to describe the participants’ demographic characteristics. Means and ranges were computed for the PROMIS Quality of Life scales and the subscales of the Caregiver Burden Scale. VHA uses were counted weekly and summed into categories. Uses were counted separately for older adults and for support persons and then combined to show the total use in each category. The interview transcripts were independently coded by three research team members. After independent coding, team members met several times to compare codes and develop themes. A low-inference content analysis guided the team’s analysis of the interview data. Low-inference descriptors involve using concrete terms, including verbatim statements of what participants said, instead of the researchers’ reconstructions about what participants said [32]. The goal in this study was to elicit data to inform future studies. 

## 3. Results

### 3.1. Sample Characteristics

Ten dyads were recruited to participate in the study. However, despite numerous attempts, we were unable to enroll the support person for one older adult, resulting in a final sample of 10 older adults and nine support persons. The sample was skewed with female participants who were White, non-Hispanic, and well-educated (Table 1). The older adults had a mean age of 75 years, and the support persons were 53 years old on average. The older adults’ chronic conditions were self-reported and included obesity, hypertension, type 2 diabetes, depression, atrial fibrillation, fibromyalgia, and a history of stroke. However, most participants reported a good quality of life. Their income levels were assessed using the Internal Revenue Service (IRS) tax-level brackets, wherein the lowest level is $18,650 or less and the highest is $233,351 or more. We selected this measure of income, because it provided a general idea of the participants’ income levels without being overly intrusive as to the exact income. The income levels for both the older adults and their support persons spanned the four levels, but most participants (80%) had incomes above the lowest bracket. Nearly all participants (90%) had a smartphone or access to a computer, and all had wireless internet services. Some participants also had smart TVs, tablets, and streaming devices, such as a Roku. 

The scales used to measure the quality of life among the older adult participants were under the umbrella of PROMIS Health: Global Physical Health, Global Mental Health, Personal Well-Being Index, Who-5 Well Being Index, and the Short Flourishing Scale (Table 2). The mean score on the Global Physical Health scale indicated that older adult participants had positive physical health (mean score = 42.00). The Global Mental Health scale scores (mean = 47.01) indicated that participants had positive mental health. Participants’ responses to the Personal Well-Being Index were varied, but most had high scores (mean = 76.94) that aligned with positive well-being. The results from the WHO-5 Well-Being Index (mean = 59.60) were similar and indicated positive well-being for the majority of the older adult participants. The responses to the Short Flourishing Scale (mean = 42.88) also reflected positive scores. In summary, taken together, the PROMIS Health scales revealed the older adult sample had positive physical and mental health and good quality of life.

The support persons included family members (*n* = 6), most frequently a son or daughter, and neighbors (*n* = 3). The most frequent assistance provided by the support persons to the participants was transportation, shopping, and cooking. The mean scores on the subscales of the Caregiver Burden Scale were all low: the Patient Needs Domain, (M = 3.88), the Caregiver Tasks Domain, (M = 2.88), and the Caregiver Burden Domain (M = 7.63) (Table 3). The low scores indicated that the older adults had low caregiving needs and that the support persons had low caregiver burdens. 

### 3.2. Quantitative VHA Use Findings

The most frequent VHA uses were requesting information, listening to music, obtaining weather forecasts, and enjoying other types of entertainment (e.g., jokes and podcasts). Older adults and their support persons used the VHAs in similar ways; however, older adults used it to listen to music more often than support persons, and support persons used it to gain information more often than older adults (Figure 1). As shown in Figure 1, the common broad categories of uses were: information (e.g., miscellaneous information, weather, and news); entertainment (e.g., music, jokes, and podcasts); and prompts (e.g., alarms, timers, and reminders/lists). Participants’ uses of the VHAs for information and entertainment remained stable over the eight weeks. In contrast, uses of the VHAs for calls primarily occurred in the first two weeks, with a relatively stable decline over the remaining six weeks. Participants’ uses of the VHAs for prompts varied over time, but prompts were used most frequently during the first week. 

### 3.3. Qualitative VHA Use Findings

The qualitative findings focused on concrete descriptors by older adult/support person dyad participants who used the Echo devices. Echo devices enable voice-activated, hands-free connection to the internet and provide the user with voice-output responses. The participants focused on obtaining information, entertainment, prompts, and, to a lesser extent, companionship and security. The challenges participants described focused on integrating Alexa with other tools that served similar purposes and older adults’ need for more education and training about the capabilities of VHAs.

#### 3.3.1. Information: “She’s Good for Information”

Older adults and support persons alike found they used their VHA for distinct types of information, including “*the weather forecast*” (SP104), “*geographical facts*” (SP108), and recipes, “…*I asked her to pull up recipes for chicken*” (OA101). Participants found the VHA to be knowledgeable, “*knows the answer to all of your questions*” and responsive, “*you just say delete it*”, and it is done (OA101). Some participants found the hands-free feature of the VHA to be a benefit. “*You don’t have to touch anything…you can be cooking or doing something else and interact with it that way”* (SP108).

#### 3.3.2. Entertainment: “…Just the Music, It Is Just a Gift”

Most older adults and support persons noted the VHA to be a source of entertainment. One older adult (101) shared, “*I didn’t realize how much I missed the music and she (VHA) will pull up different kinds of music you want to hear. My cat loves the classical, particularly the opera.*” A support person (104) explained, “*I know (my) mom really enjoyed being able to play music and that was very helpful…. she has that little happiness back*.” Another older adult (107) described the VHA as a resource for jokes. “*I was writing my grandson…a birthday card and he loves knock knock jokes. I asked Alexa to tell me knock knock jokes and she had them coming one right after another, and I used them*.”

#### 3.3.3. Prompts: “Reminders for What You Are Supposed to be Doing”

Many of the participants used the VHA to assist them to remember to complete various instrumental activities of daily living (IADLs), such as “*drink water*”, “*walk the dog*”, (OA103) and “*take medications*” (SP108). One support person (102) shared, “*I’ve used it to remind me of different things, …definitely medication more than anything, doctors’ appointments, [and] grocery lists*.” Another support person (108) informed us that she “*used it [VHA] for mostly simple things like setting timers… a timer for 15 min so I can play piano or set a timer for 10 min so I can cook pasta*.” 

### 3.4. Additional Qualitative Findings

#### 3.4.1. Companionship and Security: “Somebody Is Here with Me…” 

A couple of participants engaged often enough with the VHA that it became a companion for them. One older adult (105) shared: 


*“I have humanized that machine. I call her a she and a her and every morning I say, ‘Alexa what is the weather going to be like?’ And if it is dark outside, I say, ‘Well are we going to have thunderstorms?’ I carry on the conversation with her every morning and at night I always tell her goodnight. I say, ‘Alexa I’m going to bed.’ She says, ‘Have sweet dreams and have a good night.’ But I always report in every morning and every night and I just have a kinship with Alexa. She’s company. She’s another person. And you know, I know that’s a machine …[laughs] but it’s just that I feel like it’s somebody’s here with me.”*


Another participant noted, “*It feels like an extra level of security”* (OA109). The participant further stated: 

*“That was why we got it* [VHA] *in the first place. My right knee causes [me] a lot of problems …and I had fallen two or three times. [My daughter] thought if I fell and she wasn’t at home she wouldn’t even know. How would we connect? So, we were looking for something like that.*

#### 3.4.2. Understanding Its Capabilities and Changing Habits: “Getting Used to Another Device” 

Some participants found it took time to adapt to using the VHA. One older adult (101) shared, it is *“…a matter of forming habits…my habits are writing all this stuff down. I’ve got to break the habit of writing it [down]*.” Another older adult (107) shared that they still “*keep a paper calendar*” and then said that “*I understand that you can say, call these numbers if you program them in, but I have those [numbers] on my phone, so I just use my phone.*” The older adults and support persons struggled to figure out the capabilities of the different devices they owned and understanding those that did and did not work together. The VHA was no exception to this daily challenge. Participants were open to integrating another device into their daily lives. However, some shared how arduous it could be to determine what data was needed for input and the reasonable range of abilities that could be expected from this new device. A support person (102) shared, 


*“I did try to call [name] on it yesterday but it said [name] wasn’t in my contacts and I didn’t know what I was supposed to do to get her in my contacts. So, I said oh I will just use the phone then, so I think it sent her a text.”*


Another participant (SP110) commented that “*one of the challenges of technology …with an older person is having to keep up with whatever device it is that you’re using.”* As one older adult (101) described, maximizing the capabilities of the VHAs is a process: “*Once I can really get into it, I think I will have found …more that I am able to do with it.”* Older adults also suggested ways to improve education to use the VHA, stating that the manual was difficult to understand. The suggested instructional methods included more one-on-one education about the VHA, including (OA103) *“… a series of lesson plans …”*


#### 3.4.3. Suggested Future Applications

*“…like emergency calls”.* Participants suggested additional VHA features that would be useful. 


*The number one thing I could think of is that if you could call them through Alexa, the problem is if they are somewhere else in the house, they don’t hear it, [perhaps] if it were on them…. [In addition], if you can somehow see around the room and see if the person is there and if they are moving that would [be a] benefit.”*


Several support persons wanted the VHA to provide information about the older adults’ activities and statuses. For example, one support person (104) shared:
“…it would be nice to know … if mom had talked to Alexa that day. Did mom check in? …if I were using it for medications … mom would say, “Yeah Alexa I took my medicine this morning,” … and she could record that and I would just know that she had checked in with Alexa ...”

Other participants wanted the VHA to have more medical knowledge. One support person (104) stated:
“We tried to get her to talk to us about opiates, she could … readily pick up Wikipedia pages information. We would say tell us about—[she would reply] … I do not know or …she could not just tell us things about that. We had to know the right question to ask.”

Another support person (106) stated:
“There are some things that she [Alexa] does not know, especially in the medical field…. they change my medicine quite often and … I’ve asked her what the side effects on this kind of medicine are and … she didn’t have it. Now it could be a new medicine and it hasn’t gotten out, but in the medical field she doesn’t know the side effects of a lot of medicine.”

Another support person (106) commented that Alexa could have more health-promoting information. “[They could] *program her with more information, healthy foods to eat, stuff to stay away from, things she (the OA) should not be eating*.”

## 4. Discussion

There is a rapid societal uptake in VHA use. Over 15 million Echo smart speakers were purchased in the fourth quarter of 2019, and the 100,000 products that are compatible with Echo is growing each day [33], including many that can facilitate aging in place. Thus, understanding how to best use low-cost, commercially available VHAs and their compatible products to promote aging in place may realize many positive, cost-effective benefits for the health and well-being of the globally expanding older adult population. In this study, we described the use and perceived usefulness of VHAs (Amazon Echo products/Alexa) among ten older adult and support person dyads. Quantitative data, analyzed by part of the team (CC, LC, IS, and TN), and qualitative data, analyzed by the other team members (CVS, OO, and PW), converged to describe similar VHA uses. The range of uses described by the dyads in our study included essential and nonessential IADLs. The primary ways both groups used the VHAs were for information, entertainment, and prompts. Similar common uses have been reported by others [16,17,34,35,36,37]. The scale of use is dependent on the users knowing the VHA capabilities so that they canHA identify which ones they want to learn and use. Desired uses often require enabling a skill, inputting data, or linking the VHA to another device. For example, VHAs can respond to requests to call specific contacts, but first, the contacts need to be entered. Similarly, calls to the VHA can be forwarded to a smartphone, but the app must be downloaded on the call recipient’s phone. Using the VHA for medication reminders, calendar appointments, and scheduled check-ins with support persons requires data entry and/or enabling certain skills. 

Participants’ perceptions of the VHA’s usefulness were connected to the skills that they used. They found the VHA to be useful for information, entertainment, and prompts, such as timers and reminders. Benefits included the hands-free feature of the VHA, which has been documented by others [14,17,35]. Participants also reported that the VHA provided an added level of security by allowing older adults to contact someone in an emergency and providing a convenient way for support persons to check on older adults. Other studies have also documented that VHAs provide some users with a sense of safety or security [15,16]. Some literature [12,17] and the media describe privacy and security as a concern, but in our study, several participants reported feeling more secure with the VHA. It is likely that the self-selected sample influenced our results. A few of the older adults in our study noted that the VHA provided companionship. Several other researchers have reported a sense of companionship as a significant benefit to older adults [12,16,35]. It appears that it might be possible for the VHA to facilitate enough interactions throughout the day that the older adult may feel less isolated, particularly if they are using the video calling feature. 

Some participants shared they desired additional capabilities, not realizing that many of those functions are already available. People can pair their smartphones to their Echo device using the Echo/Alexa app on their phone. By doing so, they can interact with Alexa on their phone. The optional “Drop In” feature allows a contact who was granted permission to start an instant conversation with the person who owns the Echo. If the Echo video device (e.g., Show and Spot) were placed so that it had a view of the room, the contact would also be able to see around the room. A support person could then see, for example, if an older adult had fallen and was still lying on the floor. Thus, some of the participants’ comments about desired features already exist. This finding indicates that participants would benefit from intermittent assessments about how they would like to use the VHA, followed by training about how to use features that would meet their needs. More initial orientation and training about the VHA’s capabilities may also enhance older adults’ and their support persons’ perceptions of the VHA’s usefulness. Koon and colleagues [17] also noted that older adults sometimes suggested ideas for future development that already existed and that some of their participants had a desire to learn more about the VHA’s capabilities but were not sure how to advance their knowledge. They recommended user-focused instructional protocols to support initial and future learning about the VHA [17]. A study conducted in Germany enabled and oriented older adult uses to 13 VHA applications [35]. Participants in their study went on to independently enable between two and 67 additional applications over several months of use. Future research is needed to identify evidence-based strategies to match the VHA skills and VHA training to participants’ needs. The literature on generational diversity reveals that each generation reflects a distinct set of values, ideas, and culture, dictating a preferred learning style [38]. In a US study conducted between 2009 to 2013, workers across four generations were assessed for and interviewed about learning styles. The “silent generation” (mean age of 71 years) described a preference for informal, repetitive, hands-on training [39].

Most of our older adult and support person participants could name unfulfilled expectations they had for the VHA and ideas for what they desired it to accomplish. Most unfulfilled expectations were health-related, ranging from desiring more accurate health and medical information to communicating with healthcare professionals using the VHA. Other researchers reported similar findings [36]. In focus groups of older adults, most participants could see how the existing and new VHA applications might benefit them now or in the future. Notably, health and medical uses were the most frequently mentioned desired uses. Our team is actively developing a health-related skill for Echo, as are others. For example, research is being conducted with software used with the VHA to provide enhanced resources, such as individually tailored nutrition guidance [40,41]. Amazon has started to engage in health insurance portability and accountability act (HIPAA)-compliant business associate agreements with some health-related organizations (e.g., health systems and insurance carriers) [42]. In general, covered health entity leaders are wary of voice-activated devices and HIPAA violations, but increased VHA use in healthcare is anticipated [43].

### Limitations

The small sample size of participants selected from one geographic area limited the feasible generalizability of this study. The inclusion of more dyads with greater diversity (e.g., gender and race) may strengthen this study’s potential generalizability. In addition, a self-selection bias was introduced, as the volunteer participants already felt comfortable testing new technology and did not have privacy concerns about using a VHA. However, we believe this is the first study to describe the uses and usefulness of VHAs among older adults and support person dyads. Midway during the exit interviews, the pandemic of COVID-19 became a reality. By necessity, the post-study interviews were switched from being conducted in-person to telephone. Comparisons between the in-person to phone interviews revealed that the phone interviews generated less quality information. In addition, the absence of visual cues via telephone could have resulted in a loss of contextual and nonverbal data, which may have compromised the rapport, probing, and, possibly, interpretation of responses. Despite these limitations, this exploratory study had multiple strengths. Most notably, the study team implemented a concurrent triangulation method that reduced the study bias by comparing and contrasting the results between the qualitative and quantitative study methods. 

## 5. Conclusions

The population of older adults is rapidly increasing and will continue to expand for the next twenty years [2]. Aging in place is associated with a higher quality of life, lower healthcare costs, and less burden on federal resources [44]. We evaluated the VHA use and usefulness among ten dyads of older adults and their designated support persons, which was consistent with recommendations to investigate technology through the feedback and insight from a sample population of older adults and their support systems [44,45]. The study findings revealed that participants used the VHA regularly over time, primarily for information, entertainment, or to receive prompts. Future desired uses were primarily focused on ways that the VHA could help them promote their health and manage their health conditions. Thus, the findings contributed knowledge about the usefulness of VHAs, generated information about improvements or innovations to VHAs to meet the specific needs of older adults and their support persons, and created preliminary data for studies that will use more rigorous designs to evaluate the effects of VHA use. Information on the aspects of the protocol that worked well and aspects that could be improved, such as additional VHA training, was also gleaned from this feasibility study. This study was unique in that, to the best of our knowledge, it was the first to prospectively explore the use and usefulness of VHAs among older adult and support persons dyads.

## Figures and Tables

**Figure 1 ijerph-18-01113-f001:**
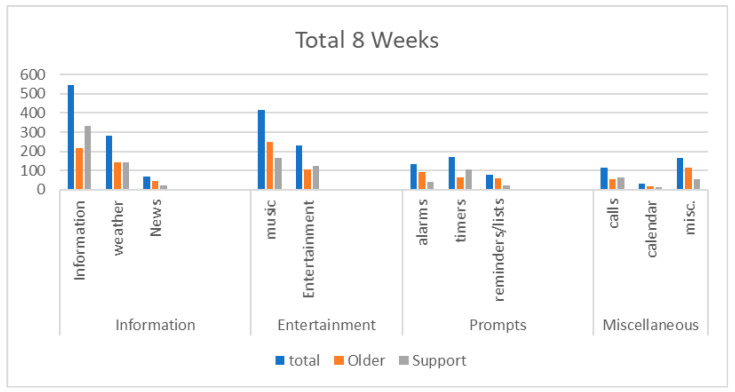
Echo uses among older adults and their support persons.

**Table 1 ijerph-18-01113-t001:** Demographic characteristics of older adults and their support persons.

Demographics ^1^	Older Adults		Support Persons	
	*n*	%	*n*	%
Sex				
Female	9	90	4	44.4
Male	1	10	5	55.5
Race/Ethnicity				
White	8	80	7	77.7
Black	2	20	1	11.1
Asian	0	0	1	11.1
Highest educational level				
Less than High School Diploma	0	0	0	0
High School Diploma/GED	2	20	2	22.2
Associates Degree/Certification	2	20	3	33.3
Bachelor’s Degree	3	30	1	11.1
Master’s and above	3	16	3	33.3
Living Alone				
Yes	8	80	2	22.2
No	2	20	7	77.7

^1^ Missing data is a result of one or more participants who declined to answer the question.

**Table 2 ijerph-18-01113-t002:** Older adult participants’ quality of life PROMIS Global Scale ^2^.

Subscale ^1^	*n*	Mean (SD)	Range
1. Global Physical Health	9	42.00 (6.14)	37.4–54.1
2. Global Mental Health	9	47.01 (9.74)	41.1–59
3. Personal Well-Being Index	9	76.94 (21.55)	34–97
4. WHO-5 ^3^ Well Being Index	9	59.60 (20.45)	36–80
5. Short Flourishing Scale	9	42.88 (10.47)	23–55

^1^ Patient-Reported outcomes Measurement Information Systems (PROMIS). ^2^ Missing data is a result of one participant declining to answer the PROMIS questions. ^3^ World Health Organization-5 Well Being Index (WHO-5).

**Table 3 ijerph-18-01113-t003:** Support persons’ Caregiver Burden Scale scores.

Subscale	*n*	Mean (SD)	Range
1. Patient Needs Domain	9	3.88 (2.30)	0–6
2. Caregiver Tasks Domain	9	2.88 (2.23)	0–5
3. Caregiver Burden Domain	9	7.63 (4.93)	0–13

## Data Availability

The data presented in this study are available on request from the corresponding author. The data are not publicly available due to the small sample size and the need to assure participant confidentiality.

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
