# Peer review of "Virtual Home Assistant Use and Perceptions of Usefulness by Older Adults and Support Person Dyads"

_ijerph, 2021, doi:10.3390/ijerph18031113_

Round 1

Reviewer 1 Report

Title: The concept “Dyads” is unclear.

Introduction:

Is it possible to give some examples related to technologies used by older adults? In addition, more detailed information related to earlier experiences of older adults and their relatives is needed to understand the context of the research.

Materials and Methods:

Who are exactly the study participants and how many? Older adults and their support persons? Please, clarify this information in different sections in the manuscript.

Ethical considerations are missing. You have mentioned written informed consent, but more detailed ethical considerations are needed. More, in Recruitment section you described inclusion criteria. What were the exclusion criteria?

Description of study protocol including measurements and interviews is a somewhat confusing. Please, clarify. For example: First, demographic information was collected from the older adult participants….. Second, …. and so on.

Analysis:

Please, clarify the quantitative analysis. What you analyzed? Descriptive statistics, frequencies, percentages, mean values, standard deviations or the sum variables or what? In addition, more detailed description of qualitative analysis is needed. More, reasons of methodological choices are needed based on previous research method literature.

Results:

Qualitative findings are narrow. There is a need for more detailed analysis. In addition, synthesis is needed to catch more comprehensive view of participants’ descriptions.

Author Response

Please see response in word document below:

Reviewer 2 Report

The paper presents an interesting study to understand the use and perception of the usefulness of Virtual home Assistants by older adults and support person dyads. Although the research seems to be very pertinent and interesting, some details need to be more explored or at least clarified to be accepted. I provided some references by the end of my report to contribute to your study.

I think the introduction is a little brief and some discussion about the problem that is being studied would be great. Moreover, describing more some provided references, such as 11,12,13,14, and 15 is important. Those works were cited but, a very general description was provided, with no reflection or contrast to the current work.

I like to point out that there is some misconception about virtual home assistants. Google Home and Amazon Echo are only devices in which run basic software to provide access to the actual home assistant via the internet. Without an internet connection, those devices are totally useless. Moreover, you can have access to them via other devices. Google home assistant is available on Android devices, such as smartphones, smart TVs, or even cars, via Android Car, which provides an interface when connected to the user smartphone. Although the Amazon device ecosystem is smaller, you can have similar options. So, I guess a little separation between these two concepts and a small explanation would clarify to a reader.

I understand the concept of triangulation design, but there are many definitions, and you could at least describe a little more your approach and provide some references to support your statement.

The paper title says the study is based on old adults, but only adults aged 65 and older were recruited. Some references point 55 as the lower bound of older adults [1], while some other points as 60 [2]. Also, there is a difference between the biological age and their own perception [3]. Many countries have their own definition. So it would be great if authors provide their references for 65 as older adult lower bound, or explain the reason for starting with 10 years older volunteers.

I apologize, but I had to do some research to understand your concept of the dyad, and borrowing from social sciences, is a group of two people, linked by some common interest (romantic, work, family, etc). So, it would be very pedagogical if you provided a very short definition. I am assuming that it is adult persons and their support ones. However, are those support persons recruited only for this study, or are they currently their supporters?

In your results, you split into four income groups. Why those ranges? Moreover, your results are presented always in absolute and percentages. It is confusing sometimes. So, since you already defined your absolute value in your Material and methods section, I think you can stick to percentages since you might be looking for a generalization. There are a lot of demographic variables, and I wonder if the authors analyzed some excerpts and to find some patterns, such as income or educational level. If yes, it would be good to present and discuss them. If Table 1 intention is to show population diversity, I think it takes a lot of space and you could shorten it a little.

In my opinion, quantitative results are presented very briefly in Figure 1. After 8 weeks of study, you could provide more numbers and discuss them. For instance, if the frequency of using the assistant decreased or increased and how that impacts the dyad.

In your conclusions, you need to spend some sentences recalling the reader about your goal. Moreover, your conclusion should provide some general achievement, which would be more than simply "...Information on aspects of the protocol that worked well and aspects that could be improved, such as additional VHA training, was also gleaned in this feasibility study..."

Finally, I wonder why Alexa was used instead of Google Assistant. Moreover, it was provided some sort of list of basic (general) actions? So, how can you know that users have the same results if they did not use the same actions? The echo spot has a small LCD. Did the users pay attention to LCD information or at least tried to interact to it? Moreover, did the authors considered any usability or user experience evaluation, before designing their own?

[1] Petry NM. A comparison of young, middle-aged, and older adult treatment-seeking pathological gamblers. Gerontologist. 2002 Feb;42(1):92-9. doi: 10.1093/geront/42.1.92. PMID: 11815703.

[2] W.H.O. Ageing and health. 2018. https://www.who.int/news-room/fact-sheets/detail/ageing-and-health

[3] William J. Chopik, Ryan H. Bremner, David J. Johnson and Hannah L. Giasson. Age Differences in Age Perceptions and Developmental Transitions. Frontieres of Psychology. Feb 2018. DOI: https://doi.org/10.3389/fpsyg.2018.00067.

Author Response

Please see comments in word document below:

Reviewer 3 Report

Digital voice-activated personal assistants — such as Amazon Echo) and Google Home— can be helpful tools to users in some everyday ways. There are quite a good bunch of comments in journals as well as web pages on this topic. Nevertheless there are quite few  research papers on the real utility of VHA in senior citizens. MIT Technology Review presented in 2017 some preliminary data on a pilot study although a final report could not be found. There are some report on the usefulness in senior with vision impairment (Ho DK. Voice-controlled virtual assistants for the older people with visual impairment. Eye (Lond). 2018;32(1):53-54. doi:10.1038/eye.2017.165) In this sese, although limited the paper present some valuable data on the use of such devices by seniors as well as main carers.

Although interesting, the paper contains only very scarce new information on the subject. At the same time, it sems to be a quite small convenience sample, as the authors have already commented.

We recommend the authors to include more papers on the revie that would enrich the discussion as well as provide some clues on areas of investigation or improvement:

  1. MARSTON, Hannah Ramsden; SAMUELS, Julie. A review of age friendly virtual assistive technologies and their effect on daily living for carers and dependent adults. En Healthcare. Multidisciplinary Digital Publishing Institute, 2019. p. 49.
  2. Rani, M. et al. “Voice Enabled Smart Home Assistant for Elderly People.” (2019)..
  3. Koon LM, McGlynn SA, Blocker KA, Rogers WA. Perceptions of Digital Assistants From Early Adopters Aged 55+. Ergonomics in Design. 2020;28(1):16-23. doi:10.1177/1064804619842501
  4. Yaghoubzadeh R., Kramer M., Pitsch K., Kopp S. (2013) Virtual Agents as Daily Assistants for Elderly or Cognitively Impaired People. In: Aylett R., Krenn B., Pelachaud C., Shimodaira H. (eds) Intelligent Virtual Agents. IVA 2013. Lecture Notes in Computer Science, vol 8108. Springer, Berlin, Heidelberg. https://doi.org/10.1007/978-3-642-40415-3_7

Author Response

(The authors gave the same response as above.)

Round 2

Reviewer 1 Report

I have checked the manuscript and you have been taken into account my previous comments to improve the manuscript. I do not have comments any more.

Reviewer 3 Report

The authors have answered my previous comments